# Phylogenetic Analyses of Sites in Different Protein Structural Environments Result in Distinct Placements of the Metazoan Root

**DOI:** 10.3390/biology9040064

**Published:** 2020-03-28

**Authors:** Akanksha Pandey, Edward L. Braun

**Affiliations:** 1Department of Biology, University of Florida, Gainesville, FL 32611, USA; aakanksha.vit@gmail.com; 2Genetics Institute, University of Florida, Gainesville, FL 32611, USA

**Keywords:** protein structure, relative solvent accessibility, non-stationary models, RY coding, heteropecilly, metazoan phylogeny, Ctenophora, Porifera

## Abstract

Phylogenomics, the use of large datasets to examine phylogeny, has revolutionized the study of evolutionary relationships. However, genome-scale data have not been able to resolve all relationships in the tree of life; this could reflect, at least in part, the poor-fit of the models used to analyze heterogeneous datasets. Some of the heterogeneity may reflect the different patterns of selection on proteins based on their structures. To test that hypothesis, we developed a pipeline to divide phylogenomic protein datasets into subsets based on secondary structure and relative solvent accessibility. We then tested whether amino acids in different structural environments had distinct signals for the topology of the deepest branches in the metazoan tree. We focused on a dataset that appeared to have a mixture of signals and we found that the most striking difference in phylogenetic signal reflected relative solvent accessibility. Analyses of exposed sites (residues located on the surface of proteins) yielded a tree that placed ctenophores sister to all other animals whereas sites buried inside proteins yielded a tree with a sponge+ctenophore clade. These differences in phylogenetic signal were not ameliorated when we conducted analyses using a set of maximum-likelihood profile mixture models. These models are very similar to the Bayesian CAT model, which has been used in many analyses of deep metazoan phylogeny. In contrast, analyses conducted after recoding amino acids to limit the impact of deviations from compositional stationarity increased the congruence in the estimates of phylogeny for exposed and buried sites; after recoding amino acid trees estimated using the exposed and buried site both supported placement of ctenophores sister to all other animals. Although the central conclusion of our analyses is that sites in different structural environments yield distinct trees when analyzed using models of protein evolution, our amino acid recoding analyses also have implications for metazoan evolution. Specifically, our results add to the evidence that ctenophores are the sister group of all other animals and they further suggest that the placozoa+cnidaria clade found in some other studies deserves more attention. Taken as a whole, these results provide striking evidence that it is necessary to achieve a better understanding of the constraints due to protein structure to improve phylogenetic estimation.

## 1. Introduction

The growing availability of very large molecular datasets has transformed the field of phylogenetics. The use of these phylogenomic datasets was suggested to “end incongruence” among phylogenetic estimates by reducing the stochastic error associated with analyses of small datasets [1]. Although phylogenomic analyses have resolved many contentious relationships [2,3,4,5], in other cases different analyses using genome-scale data have produced multiple distinct resolutions of problematic nodes, sometimes with strong support [6,7,8,9,10,11]. This suggests that analyses of these large datasets can be misled by non-historical signals that may not be as apparent in analyses of smaller datasets. Thus, rather than putting an end to incongruence, phylogenomic analyses often highlight the complexity of the phylogenetic signals present in genomic data.

The idea that non-historical signals can overwhelm historical signal in large datasets predates the phylogenomic era (e.g., the early discussion of statistical inconsistency due to long-branch attraction [12,13]). However, the availability of genome-scale data emphasizes the large number of cases where conflicting signals emerge in phylogenetic analysis. The most extreme cases correspond to those where analytical methods are subject to systematic error; these are the cases where non-historical signal overwhelms historical signal. In that part of parameter space, increasing the amount of data will cause phylogenetic methods to converge on inaccurate estimates of evolutionary history with high support [14]. Thus, phylogenomic analyses should lead either to high support for the true tree (the desired result) or to high support for an incorrect tree (if the analytical method is subject to systematic error). Cases where support is limited despite the use of large amounts of data could reflect one of two phenomena: (1) the data contains a mixture of signals; or (2) the underlying species tree contains a hard polytomy (and, therefore, historical signal is absent). If the former, the mixture of signals could be biological in nature (e.g., a mixture of histories due to reticulation) or it could reflect a mixture of historical signal and one or more non-historical signals. Understanding the distribution of historical and non-historical signal(s) in large-scale data matrices might provide insights into evolutionary processes and result in better understanding of analytical methods and their limitations with phylogenomic datasets.

The problem of systematic error in phylogenomic analyses has been addressed in several ways; one of the most popular ways to address systematic error has been the use of more complex, and presumably more realistic, models of sequence evolution. These complex models typically assume that phylogenomic data are very heterogeneous. Most of these complex models, like the CAT model [15], the Thorne-Goldman-Jones structural models [16,17,18], and structural mixture models [19], introduce this heterogeneity by assuming that distinct patterns of sequence evolution (and therefore different models) characterize different sites in multiple sequence alignments. However, there have also been some efforts to develop models that assume the heterogeneity corresponds to distinct patterns of sequence evolution on different branches in the tree of life [20,21,22]. Regardless of the specific model under consideration, the degree to which these approaches ameliorate the impact of misleading signals remains a subject of debate (e.g., see the discussion of the CAT model by Whelan and Halanych [23]). 

Identifying and examining conflicting signals within the phylogenomic data matrices has the potential to provide information about the biological basis for the heterogeneity and ultimately inform analytical approaches to deal with heterogeneous datasets. It should be possible to identify conflicting signals in phylogenomic data by dividing the data into subsets and asking whether phylogenetic analyses of those subsets support distinct trees. This raises the question of how large-scale datasets should be subdivided. Subdividing data matrices into individual loci is unlikely to be informative because individual loci are short and therefore have limited power to resolve difficult nodes [24]. Moreover, individual loci are expected to be associated with distinct gene trees due to factors like the multispecies coalescent [25,26,27]. The existence of genuine discordance among gene trees makes it necessary to sample many loci to determine whether multiple singles are present. Relatively large datasets where analyses yield trees with limited support could be explained by the presence of multiple signals. Thus, surveying publications for analyses of a relatively large dataset with low support for one (or more) specific nodes represents a strategy to identify datasets with two or more conflicting signals. Any dataset identified in this way can then be divided into two or more subsets, each of which can then be subjected to phylogenetic analyses to determine whether a mixture of conflicting signals is present.

There are many different ways to subdivide phylogenomic datasets in two or more subsets that are still large enough to overcome stochastic error. The most logical way to subdivide phylogenomic datasets involve the use of partitions that can be defined *a priori* using non-phylogenetic criteria. The simplest criteria might be functional in nature (e.g., coding vs. non-coding data [28] or genomic regions that are highly transcriptionally-active vs. those that are largely untranscribed [29]). There are two alternative hypotheses regarding the distribution of signal in any subdivided phylogenomic dataset:
**H_0_:** Conflicting signals are randomly distributed with respect to functionally defined subsets of the data. This hypothesis predicts that separate analyses of those functionally defined subsets of the data matrix will yield trees with the same topology (probably with lower support than the analysis of the complete dataset due to the smaller size of the subsets).**H_A_:** Conflicting signals are associated with functionally defined data subsets. Different subsets of the data matrix defined using functional information are associated with distinct signals (i.e., analyses of those subsets support different topologies when the subsets are analyzed separately).

Failure to reject H_0_ could reflect a genuinely random distribution of signal, failure to define the data subsets in an appropriate manner, or a definition of the subsets that reduces their size to the point where stochastic error dominates the analyses (at least for nodes that are difficult to resolve). The last issue is unlikely to be problematic because subdividing the original dataset too finely is likely to result in difficult nodes being resolved randomly with low support. The other two possibilities (truly random distribution of signal vs. use of incorrect criteria to define the data subsets) are essentially impossible to distinguish, so H_0_ cannot be corroborated in a global manner. However, it is still possible to corroborate (or falsify) H_A_ for specific ways of subdividing phylogenomic data matrices if those data subsets are sufficiently large. The most important line of evidence that the data subsets are large enough to be informative would be observing that analyses of the data subsets yield relatively high support for conflicting resolutions of a node (or multiple nodes) that have proven to be difficult to resolve in previous studies (ideally, the support observed in analyses of the subsets would be higher than the complete dataset). Efforts to do this may provide substantial information about the patterns of sequence evolution in different parts of the genome; this information is also likely to be useful for phylogenetic model development.

Coding sequences are often used to examine phylogenetic relationships deep in the tree of life (e.g., early land plant [5] or metazoan evolution [6,11] and efforts to identify to root of archaea [30] or eukaryotes [31]) and they are good candidates for this type of signal exploration. There are many biologically motivated ways to divide proteins into subsets that may reflect different selective pressures to maintain their structure and function. Ideally, the subsets should be stable over evolutionary time, and it should be practical to assign aligned sites to the subsets. Protein structure provides a good criterion for dividing the data into subsets. Protein structures diverge much more slowly than protein sequences [32,33], so it is reasonable to assume that alignments can simply be divided into structural classes that remain relatively constant over evolutionary time. Also, modern protein secondary structure prediction methods are relatively accurate (e.g., the SSPro and ACCPro programs are able to classify ~90% of residues accurately for proteins with homologs in PDB [34]), so it is possible to construct analytic pipelines to define the structural subsets. 

In this study, we examined whether different signals were associated with sites in different protein structural environments. Specifically, we subdivided globular proteins based on their structure and explored the distribution of signal supporting different placements of the root for the metazoan tree of life. Deep metazoan phylogeny has been a difficult phylogenetic problem, but there are a limited number of plausible arrangements for the major lineages (Figure 1). The traditional hypothesis (Figure 1a), which is supported by morphology [35] and some phylogenomic analyses [9,11,36,37,38], places sponges sister to all other metazoans. Many other analyses of large-scale datasets [7,39,40,41,42] support the hypothesis that ctenophores are sister to all other metazoans (Figure 1b). The third possibility, a sponge+ctenophore clade (Figure 1c), was recovered in analyses the Ryan et al. [6] genomic dataset (hereafter called the “RG dataset”) and some of the analyses in Borowiec et al. [42]. Support for the sponge+ctenophore clade in analyses of the RG dataset is limited, suggesting the RG dataset might be characterized by a mixture of conflicting signals. Our analyses revealed that the signal associated with solvent-exposed residues differed from the signal associated with buried residues; we believe that this has implications for the fit of models that are currently used for phylogenetic analyses of protein sequences.

## 2. Materials and Methods

### 2.1. Dataset

The RG dataset comprises 242 orthologous protein-coding genes extracted from genomic data for 19 taxa. Ryan et al. [6] reported that ML analyses of the RG dataset resulted in topology T3 with limited (53%) bootstrap support; the remaining 47% of the bootstrap replicates supported T2. We interpreted this limited support as evidence that the RG dataset was either too small to yield high support or that it is characterized by a mixture of signals. We used the alignments provided by Ryan et al. [6] without changes; the sequences had been aligned using CLUSTALW [44] with default parameters and poorly-aligned regions had been excluded using Gblocks [45]. We inspected all of the Ryan et al. [6] alignments visually in Geneoius v. 9.1.5 (Biomatters Ltd., Auckland, New Zealand); none of the alignments appeared problematic. We used the TopCons prediction server [46] to determine whether there were any transmembrane proteins in the RG data. This resulted in the identification of 10 transmembrane proteins (Appendix A), which were removed because our structural assignment pipeline was not appropriate for those proteins. This resulted in a 232-protein dataset with 102,000 sites and 19.8% missing data. 

We conducted BLASTP searches of UNIPROT [47] to determine the identity and function of each gene (using annotation of the human sequence for most genes). There were 22 cases where the human sequence was absent from the RG alignment; in those cases, we used the *Drosophila* sequence to identify the genes. These functional annotations are available in Appendix A. This analysis revealed that remaining proteins represent a diverse set of globular proteins without an overrepresentation of sequences from a particular protein family making it a relatively unbiased dataset. We provide all protein alignments as Nexus files with structural annotation, generated as described below, in Appendix A. We analyzed two taxon samples: (1) the full taxon sample comprising all 19 taxa in the RG dataset; and (2) a reduced taxon sample that excluded four relatively divergent outgroup taxa [two taxa in Holozoa (*Capsaspora* and *Sphaeroforma*) that fall outside of Metazoa and Choanozoa and two fungi (*Saccharomyces* and *Spizellomyces*)].

### 2.2. Structural Class Assignment

We assigned secondary structures to the MSAs using the SSpro and ACCpro programs in the SCRATCH 1D suite [48]. SSpro classifies sequences into three secondary structural classes (helix, sheet, and coil) [34]. ACCpro assigns each residue to one of the two categories: exposed (e) or buried (-) [49] with the latter defined as amino acids with <25% relative solvent accessibility. We used a weighted consensus sequence representing each protein in the RG dataset as input for SSpro and ACCpro. We generated a weighted consensus sequence for each protein using the Henikoff and Henikoff [50] method; the amino acid residue with the highest weight at each position was used in the consensus sequence. The structural data were extrapolated from the consensus sequence to the whole alignment and then the alignment written along with CHARSETS for each structural subset in a nexus file [51]. We then extracted sites of a given structural class from all the genes and created a concatenated alignment for a given structural class. The Perl program for this analysis is available from (https://github.com/aakanksha12/Structural_class_assignment_pipeline). 

### 2.3. Phylogenetic Analyses

We used RAxML v. 8.2.4 for log likelihood estimation and tree searches. We examined a set of standard empirical models (LG [52], WAG [53], VT [54], JTT-DCMUT [55] and rtREV [56]) with maximum-likelihood (ML) estimation of the equilibrium amino acid frequency parameters (e.g., the -m PROTGAMMALGX option in RAxML) and we used GTR model parameters optimized on the complete dataset (hereafter, we call the parameter estimates “grand GTR parameters”) and various subsets of the data (see below for a description of the structural partitions). Whenever we conducted analyses in RAxML using the GTR model we used GTRGAMMA (i.e., the GTR model combined with a four-category discrete approximation to the Γ distribution that describes rates across sites). We assessed the nodal support using rapid bootstrap [57] with the bootstopping criterion [58] as implemented in RaxML (i.e., the -N autoMR option).

To examine whether there were any observed differences among partitions in their signal we searched for decisive sites (cf. Kimball et al. [58]). Decisive sites are the sites with a very large impact on the likelihood given each a specific topology. To do this we calculated per site log likelihoods for each candidate topology using RaxML (via the ‘-f G’ option in the program). We calculated Δln*L* for individual sites; we viewed sites with Δln*L* > 5 standard deviations from the mean Δln*L* as decisive sites, following Kimball et al. [59].

We also conducted preliminary analyses of 232-protein dataset by dividing the alignment into structural subsets; analyses of the exposed and buried subsets resulted in two distinct trees identical to those in Figure 2. We then determined whether any specific gene strongly favored either topology using “outlier gene” analysis [60]. We found that gene 41 had a strong signal favoring the T3 in Figure 1. Unpartitioned analyses using the LG model the likelihood difference given the two trees in Figure 2 (which have topologies T2 and T3) was more than three-fold greater for gene product 41 than for any of the other proteins in the data matrix (Δ*ln*L = 106.63 favoring the buried tree for protein 41 compared to a range of Δ*ln*L = 9.39 to 28.67 for the other proteins; see Appendix A). Because protein 41 was an outlier relative to the other sequences we removed it to yield a 231-protein data matrix called the filtered Ryan genomic (FRG) dataset. 

We did not explore the basis for the unusual signal associated with gene 41 any further. The most straightforward reason for the unusual signal is that gene 41 was an incorrect orthology call. Indeed, the goal of this outlier gene analysis was to identify any cases of hidden paralogy in the Ryan et al. [6] gene set. However, there are two alternatives to hidden paralogy that could explain the unusual signal. The first alternative is that the gene tree associated with gene 41 might differ from the species tree due to the multispecies coalescent [25,26,27], although it is unclear whether the branches at the base of the metazoan species tree are short enough for discordant gene trees to be common. The second alternative is that there could be some unknown source of bias associated with gene 41. Regardless, removing gene 41 did not have a substantive impact on the trees recovered after dividing sites into subsets based on relative solvent accessibility. Analysis of exposed sites from the 232-protein dataset using the LG+Γ model resulted in a tree with a topology identical to that recovered when exposed sites from the FRG dataset were analyzed (Figure 2). Specifically, we found 97% bootstrap support for T2 and 99% support for the placozoa+cnidaria clade. Buried sites from the 232-protein dataset yielded a tree with one rearrangement relative to the buried FRG tree when analyzed using the same method; the rearrangement was in the outgroup. The ingroup topology for the 232-protein buried tree was identical to that for the FRG dataset; bootstrap support for T3 was 87% and support for the cnidaria+bilateria clade was 55%. Both trees are available in Appendix A. 

### 2.4. Model Estimation and Model Comparisons

We obtained ML estimates of the model parameters (amino acid exchangeabilities and amino acid frequencies) for each structural class using RAxML [61]. Our approach relaxes the notion that all sites evolve following the same Markovian process, but it still assumes that all the sites in the same subset of the MSA follow the same stationary and homogeneous Markov process (i.e., each subset has its own set of GTR parameter estimates). We examined the following classes:Two relative solvent accessibility (RSA) based classes (EXPOSED and BURIED).Three secondary structure-based classes (HELIX, SHEET, and COIL).Six classes, combining RSA and secondary structure (HELIX_EXP, HELIX_BUR, SHEET_EXP, SHEET_BUR, COIL_EXP, and COIL_BUR).

We estimated the GTR model parameters for each structural class using the -f e option in RAxML and then performed a tree search. We used the ML tree from Ryan et al. [6] as a starting tree and estimated the model parameters. If the best tree obtained using the estimated parameters differed from the starting tree, we re-estimated model parameters using the new tree, iterating this procedure until the input and output tree converged (cf. Le and Gascuel [18]). Hereafter, the model parameters optimized for each structural class are further referred to as structure-based model estimates; the parameter estimates are available in Appendix A.

We used multidimensional scaling in R [62] to visualize the differences among estimates of exchange rate parameters obtained from the structural partitions as well as the standard empirical models. The exchangeability matrices for time-reversible models or protein sequence evolution models have 190 elements so we treated each model as a 190-element vector. These vectors were normalized so the elements summed to one and they were used to create a matrix of Euclidean distances among the models. Then we used R cmdscale function to reduce this matrix to two dimensions (link to the R script: https://github.com/aakanksha12/Multidimensional-Scaling/). 

To test whether the different rate matrix parameter estimates might differ due to sampling variance alone we randomly sampled sites ranging from 500–55,000 from the original concatenated dataset and estimated the GTR exchangeability parameters on these samples. We then calculated the Euclidean distance between the GTR exchangeability parameters estimated using the complete concatenated dataset (the grand GTR parameters) and those estimated using the randomly sampled sites. These distances were then compared to the distance between GTR parameters estimated using each structural class. 

We assessed the accuracy of our structural assignments using estimated equilibrium amino acid frequencies (following convention, we use π*_i_* for the equilibrium frequency of the *i*th amino acid). If our structural assignments are accurate, we expect: (1) polar residues will dominate the exposed sites and hydrophobic residues will dominate the buried sites [63]; and (2) equilibrium amino acid frequencies for each secondary structural class will be correlated with amino acid propensities for each secondary structure. To examine these predictions, we calculated ∆ values (π*_i_* for each structural model minus π*_i_* for all sites) and examined the correlation between those ∆ values and the relevant amino acid characteristic. Polarity (using the Grantham [64] polarity scale) and ∆-exposed were positively correlated (r^2^ = 0.7555) whereas polarity and ∆-buried were negatively correlated (r^2^ = 0.7551). Secondary structural ∆ values were positively correlated with the Fujiwara et al. [65] helix and sheet propensities (r^2^ = 0.8006 for helices and r^2^ = 0.9334 for sheets). ∆-coil was positively correlated with the Chou and Fasman [66] coil parameter (r^2^ = 0.6771). Glycine and proline are expected to be abundant in coils and those amino acids were overrepresented in our coil model; the coil model had π*_G_* = 0.0905 and π*_P_* = 0.0678 (compared to π*_G_* = 0.0516 and π*_P_* = 0.0358 for all sites). Estimated π*_i_* values along with the polarity and secondary structure propensities are provided in Appendix A. The expected correlations between ∆ values and the relevant amino acid properties were found in all analyses; those results indicate the alignment masking used by Ryan et al. [6] did not interfere with our structural assignments.

### 2.5. Analyses Using Site-Heterogeneous CAT-Type Models

We wanted to determine whether the topological differences between trees estimated using exposed and buried residues could be reduced using models that incorporate heterogeneity among sites in the sets of amino acids permitted at each site (typically called ‘profiles’). To do this, we used the CAT-type models [67] implemented in IQ-TREE v. 1.5.3 [68]. These models differ from the Lartillot and Philippe [15] CAT model in their details, although they are closely related (see below). For these analyses, we used all profile mixture classes (C10-C60) with the I+G4+FO options. We analyzed the exposed and buried alignments and assessed nodal support using the ultrafast bootstrap [69] with 1000 replicates (-bb 1000). Because we used the ultrafast bootstrap for these analyses and the rapid bootstrap in the RAxML analyses, we repeated the GTR bootstrap analyses in IQ-TREE (using the rate matrices estimated by RAxML).

The difference between the IQ-TREE CAT-type models and the Lartillot and Philippe [15] CAT model involves the calculation of the amino acid frequency profiles. Briefly, the likelihood given CAT-type models is calculated in the same way other models of sequence evolution (reviewed in *section 8.2* of Warnow [70]). The probability of finding amino acid *j* after time *t* if amino acid *i* is ancestral is calculated by exponentiating the product of an instantaneous rate matrix (often called ***Q***) and *t*. Felsenstein’s tree-pruning algorithm [71] is then used to calculate the probability of each site pattern given the complete tree with the branch lengths. For time-reversible models, ***Q*** matrices are populated by setting the off-diagonal elements to *q_ij_* = *r_ij_*π*_j_* (*r_ij_* values are amino acid exchangeabilities; as above, π*_j_* is the equilibrium frequency of amino acid *j*). The diagonal elements of ***Q*** are the negative sums of the off-diagonal elements on the same row. Then the ***Q*** matrix is normalized to allow *t* to be expressed as substitutions per site.

The core feature of CAT-type models is their use of a mixture of ***Q*** matrices (each with a specific weight) generated using single exchangeability matrix ***R*** (which contains the *r_ij_* values) and a set of profiles (vectors of π values) [15,67]. The idea underlying CAT-type models is the observation that many sites in proteins only accept a subset of the 20 amino acids, an idea corroborated by many recent mutagenesis studies (see Melamed et al. [72] for an example that also includes a comparison of mutagenesis-derived and evolutionary profiles). CAT-type models accomplish using profiles, which can be “narrow” (i.e., they can assign high π values to a few amino acids and low π values for the remaining amino acids) or “wide” (i.e., they assign most amino acids similar π values). The CAT model was described in a Bayesian framework where the profiles are sampled from a Dirichlet process prior [15]. In contrast, the profiles for the CAT-type models implemented in IQ-TREE are fixed profiles estimated from a large training dataset [67]. To acknowledge the use of fixed profiles, we call these CAT-type models “LGL models” to indicate the authors of the paper that described the profiles (Le, Gascuel, and Lartillot).

Profile “width” can be assessed using their effective alphabet size (cf. equation 5 from Pollock et al. [73]). For example, the narrowest LGL profile (effective alphabet size of 1.93) models conserved tryptophan residues; it has π*_W_* = 0.864 and low π values for the other amino acids (16 of the 20 amino acids have π < 0.01; Appendix A). The effective alphabet sizes of LGL profiles range from 4.71 to 16.23 (median = 10.17) for C10 and from 1.93 to 17.39 (median = 8.68) for C60. For comparison, the effective alphabet size implied by the LG model [52] frequencies is 18.04 (Appendix A). CAT-type models are mixture models, so individual sites are not assigned to any specific profile. Instead, site pattern likelihoods are calculated as the weighted sum over all profiles.

Although the profiles are the focus of CAT-type models, it seems clear that the ***R*** matrix might have substantial impact on the results of analyses as the profiles. The simplest implementation of CAT in IQ-TREE uses equal values for all ***R*** matrix elements. An amino acid ***R*** matrix where all elements are equal corresponds to the 20-state F81 model [71]; thus, we use “LGL-F81 models” for these models. It is unlikely that all pairs of amino acids have equal exchangeabilities, so we also conducted CAT analyses using the exposed or buried rate matrices estimated using RaxML (i.e., we used -m <matrix>+C##+I+G4+FO in IQ-TREE; C## indicates the profile mixture). This approximates the CAT-GTR model [15]; for clarity, we call these models LGL-BUR and LGL-EXP (for buried and exposed residues, respectively).

### 2.6. Compositional Heterogeneity and Data Recoding

To examine differences among taxa in their overall amino acid composition we focused on two groups of amino acids: those encoded by GC-rich codons (G, A, R, and P) and those encoded by AT-rich codons (F, Y, M, I, N, and K) [74]. Briefly, we excluded parsimony uninformative sites, counted the numbers of amino acids in each group, and calculated the GARP/FYMINK ratio for parsimony informative sites. To complement our analyses of the GARP/FYMINK ratio we also back translated sequences and examined variation along the three axes that describe nucleotide composition (i.e., the strong-weak (GC-AT), amino-keto (AC-GT), and purine-pyrimidine (AG-CT) axes) for the parsimony informative first and second codon positions.

We also recoded the amino acid data matrix in two ways: (1) six-state Dayhoff coding [75,76] and (2) RY (binary) coding [77]. The six-state recoding of amino acids assigns the following codes to the twenty amino acids: 0 = C (cysteine); 1 = A, G, P, S, and T (small residues); 2 = N, D, E, and Q (acidic and amide residues); 3 = R, H, and K (basic residues); 4 = I, L, M, and V (large aliphatic residue); and 5 = F, W, and Y (aromatic residues). The six-state data matrix was analyzed using the MULTIGAMMA model in RaxML, with the other settings described above. Binary coding involved back translating the amino acids but coding purines as 0 and pyrimidines as 1 (nucleotides that were ambiguous at this level were coded as ‘?’). The binary data matrix was analyzed in RaxML using the BINGAMMA model after completely ambiguous columns were excluded. This operation is identical to RY coding of nucleotide matrices and it yielded a 142,358-site matrix (30.6% missing data/gaps) for buried sites and a 135,945-site matrix (37.1% missing data/gaps) for exposed sites. The recoded data, generated using a Perl program available from https://github.com/aakanksha12/recodeAA, are provided in Appendix A.

## 3. Results

### 3.1. Sites in Distinct Structural Environments Have Different Signals

Different structural classes were associated with different phylogenetic signals based on analyses using standard empirical models. The most obvious difference was evident when we divided the FRG dataset using relative solvent accessibility (Figure 1). Analyses of solvent-exposed sites placed ctenophores sister to all other metazoan (topology T2 in Figure 1), whereas analyses of the buried sites recovered a sponge+ctenophore clade (topology T3 in Figure 1). The best-fitting standard empirical model was LG [52], although our results were robust to the use of different models for ML analyses (Table 1). The position of placozoa also differed in trees generated by analyses of exposed and buried sites, although support for the placozoa+bilateria clade in the analyses of buried sites was limited (<70%). The results of analyses using empirical models did not change when using the 20-state general time reversible (GTR) model, which had a better fit to both of the structurally-defined subsets of the FRG data than the LG model despite the large number of free parameters that must be optimized for that model.

Further subdivision of the FRG data using secondary structure information (helix, sheet, and coil) generally had limited impact on topology (Figure 3). Bootstrap support for the focal node was limited in most analyses. Subdividing the helix and coil sites into exposed vs. buried subsets still revealed the different signals evident when the exposed and buried sites were defined globally (Appendix A); the exception was the analyses of buried sheet residues, which placed ctenophores sister to all other metazoan (unlike analyses of the other buried residues). However, all analyses of sheet residues, even when those sites were divided into solvent-exposed vs. buried sites, resulted in an unexpected clade comprising sponges and placozoa (Appendix A). Overall, these results indicate that the data subsets with and the signals in the FRG dataset support either tree T2 or T3 (Figure 2) and that the largest difference in signal was evident when sites are divided into those exposed to solvent vs. those buried in the interior of the proteins.

### 3.2. Decisive Sites Reveal Conflicts within Each Structural Class

Strong phylogenetic signals are often limited to a subset of genes [60,79,80] or even to specific sites within genes [59,81]. Examining the sites with the strongest phylogenetic signals can provide a way to determine the distribution and amount of conflict within the data subsets. We examined the number of “decisive sites” (sites that strongly support either of two trees in Figure 2) in the exposed and buried classes, and found significant differences (*p* < 0.02, Fisher’s exact test) in the numbers of decisive sites that correspond to solvent-exposed sites vs. those associated with buried residues (Table 2). We also examined the concentration of decisive sites within different genes in exposed and buried residues and found that they were uniformly distributed among the genes. These results indicate that differences in signal in different parts of the FRG dataset that can be detected in comparisons of solvent-exposed vs. buried sites does not reflect an unusual concentration of decisive sites in any particular gene within the FRG dataset; instead, the contrasting signals appear to be a more universal feature of analyses focused on sites in the two different structural environments.

### 3.3. Reduced Outgroup Sampling also Reduces Some Differences in Signal

Highly divergent outgroup taxa are known to affect phylogenetic inference. Several studies focused on the position of the metazoan root have analyzed reduced taxon sets that exclude divergent outgroups [6,36]. We conducted the same experiment by deleting the four divergent outgroups in the complete dataset [two fungi (*Saccharomyces* and *Spizellomyces*) and two taxa from the classes Mesomycetozoea (*Sphaeroforma*) and Filasterea (*Capsaspora*)] from the FRG dataset. This limits the outgroups to Choanozoa, the sister group of metazoa [78,82,83]. Analyses of exposed and buried residues using the FRG dataset with the reduced taxon sample largely converged on a single basal topology (T2) (Figure 3).

Although the position of the root was more consistent in analyses of the reduced taxon sample than it was in the analyses all taxa (Figure 3), divergent outgroup removal does not eliminate the distinct signals associated with exposed and buried sites. The position of cnidaria depended on relative solvent accessibility, even when the reduced taxon sample was analyzed (Appendix A). Most exposed site analyses of placed cnidaria sister to placozoa (with 72% bootstrap support in the analysis of all exposed sites) but buried site analyses typically revealed a cnidaria+bilateria clade (albeit with limited support in many cases; Appendix A). We observed a maximum bootstrap support of 80% for the cnidaria+bilateria clade (in the buried sites with a sheet secondary structure). However, the analysis of all buried sites has much lower support (55%); this is likely to reflect conflict within the buried sites since the buried coil sites actually yield the placozoa+cnidaria clade with very limited support; Appendix A).

Understanding the behavior of analyses using these reduced taxon samples is challenging. We conducted these taxon sampling experiments to test the hypothesis [36] that including divergent outgroups might improve phylogenetic estimation. However, including the divergent outgroups actually reduces the length of the branch separating the ingroup and outgroup (Appendix A) and it might therefore reduce the potential for long-branch attraction. Moreover, we only observed increased congruence between exposed and buried sites regarding the position of the root; the position of cnidaria remained incongruent in analyses of the reduced taxon sample. Overall, our primary conclusion is that analyses of the reduced taxon sample further underline the complexity of the signals in the FRG dataset and emphasize the differences among the structural environments in their phylogenetic signal.

### 3.4. Sites in Different Structural Environments Exhibit Distinct Patterns of Sequence Evolution

In standard empirical models (e.g., the LG, WAG, and Dayhoff models) and in the GTR model with parameters optimized for a specific large-scale protein alignment, the rate matrices reflect the patterns of sequence evolution across all structural classes. Thus, the values in the empirical rate matrix might not reflect estimates from structurally divided data. We next tested if patterns of sequence evolution in each structural environment differed. When we examined differences among models using multidimensional scaling, the strongest separation among models was related to the best models for the two solvent accessibility classes (Figure 4). Standard models formed a cluster closer to models estimated using buried residues (Figure 4 and Appendix A).

The various structural subsets have different numbers of sites, and the GTR model for amino acids has a large number of free parameters, raising the question of whether the observed differences in exchange rate parameter estimates simply reflect sampling error. Yet the models appear to cluster in a manner that is correlated with structural class in multidimensional scaling space (e.g., note that buried sites and solvent-exposed sites cluster in Figure 4 even when they are further subdivided into independent sets of helical, sheet, and coil sites), suggesting that sampling error is unlikely to explain the observed difference. Nevertheless, we also sampled sites from the FRG dataset randomly (i.e., without respect to structure) to generate datasets comparable in size to the structurally defined subsets and estimated model parameters on these random samples. Since the sites were sampled randomly, estimates of GTR model parameters should converge on the values estimated from the dataset as a whole, assuming that the number of sites that were sampled is sufficient to overcome sampling variance. We assessed the distance between the GTR model parameter estimates for the complete FRG dataset to those of randomly selected sites. We found that this distance rapidly decreased as the number of sites in the random sample increased (Appendix A); the distances between the “global” model based on the complete FRG dataset and the estimates based on each structural subset are much greater than the distances expected based on sampling variance. This demonstrates that the differences in parameter estimates of structural classes differ much more than expected based on sampling variance.

### 3.5. Site-Heterogeneous Profile Mixture Models can Yield Surprising Topological Changes

Site heterogeneous models, like the CAT model [15], represent another way to accommodate heterogeneity in the evolutionary process. As described in Section 2.5 of the Materials and Methods, CAT models assume a mixture of distinct evolutionary processes that differ in the equilibrium frequencies of the 20 amino acids. To complement the ML analyses using empirical models and the GTR model, we analyzed exposed and buried sites using site-heterogeneous models implemented in IQ-TREE [68]. These models [67], which we call LGL models, are similar to the Bayesian CAT model but they can be used in an ML framework. We analyzed the full taxon set and the reduced outgroup taxon set using all six LGL models (C10 through C60). If accommodating variation among sites in their propensity to accept specific amino acid substitutions is necessary to obtain accurate estimates of phylogeny, then analyses using the LGL models should result in trees based on the two structural subsets (exposed vs. buried) converging on the same topology.

Most analyses of the exposed and buried sites from the full taxon set using the LGL model with various number of profile mixtures and the F81 exchangeability matrix (see Section 2.5 in Materials and Methods for details) converged on a single tree T3 (Table 3). Analyses of exposed residues using three profile mixtures (C10, C30, and C50) placed ctenophores sister to other metazoans. However, two of these analyses also resulted in a tree with the sponge+placozoa clade (Pori+Pla in Table 3). The sponge+placozoa clade was only recovered in analyses that included all outgroups. In contrast to analyses using the GTR models, analyses of the reduced taxon sample using the CAT-F81 models resulted in different basal topologies for exposed and buried classes (T2 and T3 respectively; Table 3). Overall, it is important to recognize that analyses that used LGL-F81 models sometimes resulted in a surprising clade (sponge+placozoa) and that analyses using the LGL-F81 model did not provide clear evidence for increased congruence between the exposed and buried sites.

Analyses of buried residues using the LGL-F81 models also resulted in non-monophyly of Deuterostomia; all profile mixtures placed the sea urchin *Strongylocentrotus purpuratus* sister to a clade comprising chordates along with all protostomes in the taxon sample, in contrast to results obtained using the GTR model (Figure 5). To test whether any of the observed topologies reflected differences between models or the programs IQ-TREE and RAxML (to determine whether our results reflected differences between the search or numerical optimization routines in those programs rather than the use of the LGL-F81 models); we confirmed that analyses using the site-homogeneous GTR model in IQ-TREE yielded the same results as analyses in RAxML (Figure 5, Appendix A). Although the most complex LGL profile mixture (C60) had the best fit to the data based on the AIC*_c_* (Table 3) other results that emerged from the use of the LGL models could indicate over fitting of the model. Specifically, the LGL-F81 models with a larger number of profile mixtures also had zero or near zero weights for some mixture components (Appendix A). Regardless of the details, it seems clear that the unexpected signal in the buried residues (deuterostome non-monophyly) revealed by LGL-F81 models is very likely to be non-historical.

There is evidence that the F81 exchangeability matrix is problematic for CAT models [23] so we repeated our analyses using an ***R*** matrix with unequal values. We combined the GTR rate matrices estimated for the buried and exposed sites with the Le et al. [67] profiles. These models, which we call LGL-BUR and LGL-EXP, should be similar to the CAT-GTR model described by Lartillot and Philippe [15] but they can be used in an ML framework. The LGL-BUR/EXP models had a much better fit to both datasets than the LGL-F81 models (compare the AIC*_c_* values in Table 3 and Table 4). As we observed for the LGL-F81, some mixture weights for the LGL-BUR/EXP models were zero or near zero. One major difference between the LGL-F81 and LGL-BUR/EXP models was the estimated FO mixture component weight. The FO mixture component amino acid frequencies are estimated from the data, unlike the amino acid frequencies fixed profiles that were estimated from training data. For the LGL-BUR/EXP models the FO mixture weight was as high as 0.3637 (see Table 4 for all values); this was much higher than the FO weight for the LGL-F81 analyses (which ranged from 0.0169 to 0.0746; see Appendix A).

Overall, the topologies of trees recovered using the LGL-BUR/EXP models were much more stable than those conducted using LGL-F81. All analyses of exposed and buried sites using the LGL-BUR/EXP models supported the cnidaria+placozoa clade, often with relatively high bootstrap support (Table 4). Likewise, none of the unexpected relationships, like the sponge+placozoa clade or the chordate+protostome clade (i.e., the clade that implies deuterostome non-monophyly), were recovered in analyses of the complete taxon sample that used LGL-BUR/EXP. Most analyses of exposed sites recovered T2 and all analyses of the buried sites recovered T3 (Table 4). For the exposed sites, the only analysis of the complete taxon sample that failed to recover T2 was the LGL-EXP-C60. Although LGL-EXP-C60 had the best-fit to the data based on the AIC*_c_* analyses using that model resulted in the T3 topology, albeit with limited support (63%; Table 4).

The results were more complex for the reduced taxon sample. All LGL-EXP analyses resulted in T2 (Table 4) whereas the LGL-BUR analyses were split between T2 and T3. More specifically, analyses of buried sites yielded T2 when models with a limited number of mixture components (C10 and C30) were used but analyses using more complex LGL-BUR models (C50 and C60) resulted in T3 (Table 4). The optimal trees for two LGL-BUR models of intermediate complexity (C20 and C40) were T3, but the sponge+ctenophore clade had very low support and the bootstrap consensus tree topology was T2. All LGL-EXP analyses yielded strong support for deuterostome monophyly (Figure 5). In contrast, the results of analyses using buried sites and the LGL-BUR model were more complex. The deuterostome clade was recovered in analyses that used all outgroups, albeit with moderate support (Figure 5). On the other hand, most of the analyses that used the reduced taxon sample yielded deuterostome non-monophyly, although bootstrap support for the unexpected chordate+protostome clade was also relatively low (Figure 5).

### 3.6. Binary Recoding Eliminates the Observed Differences in Signal

The GARP/FYMINK ratio, a metric of amino acid composition that correlates with genomic GC-content, differed among taxa for both exposed and buried sites and the mean GARP/FYMINK ratios for the two structural classes also differed (Figure 6a). However, the pattern of variation among taxa in the ratio relative to the means was virtually identical for both structural environments. Specifically, the GARP/FYMINK ratio was lower than the mean for non-bilaterian animals and higher than the mean in most non-metazoan outgroup taxa. Despite the evidence for variation among taxa there was no evidence for large-scale convergence between either ctenophores or sponges and the outgroups (Figure 6a). Although a simple pattern of convergence was not evident, amino acid compositional variation does violate the assumptions of time reversible models and this could have an impact on phylogenetic inference. Thus, limiting compositional variation has the potential to improve our estimates of phylogeny. We examined variation in nucleotide composition among taxa along the three possible axes of nucleotide composition (Figure 6b). Since the largest degree of variation was on the strong-weak axis we reasoned that RY coding might reduce the impact of compositional variation.

Phylogenetic analyses of exposed and buried sites after binary coding resulted in identical trees that placed ctenophores sister to all other animals with strong (≥88%) bootstrap support (Figure 6c). Analyses of the exposed sites after six-state Dayhoff recoding resulted in the same topology, also with high support for placing ctenophores sister to other metazoa (Figure 6c). Similarly, the analyses of the buried sites after six-state coding also resulted in a tree with ctenophores sister to all other animals, albeit with limited (48%) bootstrap support (Appendix A). The buried Dayhoff recoded tree also had a sponge+placozoa clade, although that unexpected clade had very low (28%) bootstrap support (Appendix A). Regardless of the details, applying either recoding method to the buried sites results in congruent placements of the metazoan root, specifically the T2 topology, when either the exposed or buried subsets of the data were analyzed.

## 4. Discussion

Analysis of the FRG dataset conducted after separating sites into two subsets using relative solvent accessibility resulted in two different tree topologies (e.g., Figure 2). We also observed significant differences in the numbers of decisive sites that favor distinct placements of the metazoan root in the exposed and buried site subsets (*p* < 0.02, Fisher’s exact test; data presented in Table 2). These results strongly indicate that conflicting phylogenetic signals are non-randomly distributed in parts of the dataset that can be defined using protein structure. Although the other strategies we used to subdivide the data (i.e., dividing aligned sites based on their secondary structure or using a combination of secondary structure and relative solvent accessibility) revealed some differences in signal, the topological differences observed in those cases were poorly supported relative to the differences apparent when solvent-exposed sites was compared buried sites. These results corroborate the alternative hypothesis we articulated in the introduction, but only for exposed vs. buried sites. The different signals in exposed and buried sites were evident regardless of whether the phylogenetic inference was conducted using standard empirical models or with GTR model parameters optimized on each structural subset. Moreover, the distinct signals persisted when the exposed and buried data subsets were analyzed using LGL models, which are site-heterogeneous CAT-type models. However, amino acid recoding reduced the differences in signal for exposed and buried sites and, in the case of analyses using binary coding, revealed support for a single tree that placed the metazoan root between ctenophores and all other animals (i.e., topology T3) with a relatively high degree of support (Figure 6c).

### 4.1. Different Models for Different Structural Environments

The obvious explanation for the conflicting signals in each structural class, especially the strong difference between the exposed and buried signals, is that the sites in each of these classes exhibit different patterns of sequence evolution. The poor fit of standard empirical models to the data could then result in an incorrect inference. This could result in the observed difference in signal, with one structural subset yielding the correct tree while analyses of the other subset result in an incorrect estimate of phylogeny. Alternatively, it is possible that analyses of both exposed and buried sites yield different topologies, both of which are inaccurate. GTR model parameter estimates for each class certainly indicate that the patterns of sequence evolution are different in each structural environment (Figure 4 and Appendix A). However, conducting analyses using GTR model parameters optimized on each class did not cause analyses in the two classes to converge on a single topology (in fact, the topologies were unchanged relative to those that emerged in the analyses that used standard empirical models). Finally, we note one intriguing aspect of our model comparisons: the rate matrix parameters for most empirical models lie closer to the models based on buried sites (this was true for all empirical models trained on diverse datasets; i.e., all empirical models except those trained using the more limited organellar or viral datasets).

The basis for the differences we observed among the structural classes in their patterns of evolution almost certainly reflects differences in the nature of the purifying selection on sites in different structural environments. It is well known that protein evolution is heterogeneous and depends on the site-specific biochemical constraints like structure, dynamics, and biochemical functions [84]. There is substantial evidence that, depending on their position and role in the overall conformation and function sites will only accept a subset of the 20 amino acids, with all other possibilities being selected against [85]. For example, in the case of relative solvent accessibility, the sites that are exposed to an aqueous environment will include more polar residues whereas buried sites which form the cores of proteins and are inaccessible to solvent, will include more non-polar residues and be more resistant to changes in side chain volume [86]. Similar differences among the major secondary structural classes have also been appreciated for some time [16,17,18], although the differences among the secondary structure classes does not appear to be as extreme as the differences based on solvent accessibility. Overall, these differences in evolutionary patterns among the structural classes can lead to different signals in each structural class.

Models of sequence evolution that incorporate protein structure have also been proposed in the context of phylogenetics. For example, Goldman et al. [16] used a hidden Markov model approach to assign distinct rate matrices to sites based on in secondary structure and solvent accessibility (where secondary structure and solvent accessibility are the hidden states). Le and Gascuel [18] developed a mixture model that uses available protein structure annotations. Both of these studies revealed that efforts to acknowledge the different structural classes for sites in protein multiple sequence alignments can have a major impact on model fit, measured using the improvement in log likelihood values. However, both of those methods estimate a tree topology for the complete alignment; our approach of dividing proteins into structural classes follows the same basic idea but shifts the focus to study the phylogenetic signals in these distinct structural classes separately.

The primary focus of this study was testing whether different signals emerge in analyses of sites in distinct structural environments rather than phylogenetic inference *per se*. This led us to analyze structurally defined subsets of the data. In principle, analyzing all sites in a sequence alignment using a model that appropriately accommodates heterogeneity should yield the best estimate of phylogeny. However, that approach is not ideal for examining conflicting signals or asking whether the signals are non-randomly distributed. Observing different topologies when data are analyzed using models that are structure-aware vs. models that are not structure-aware could result from the existence of distinct signals for sites in different structural environments. Alternatively, it could reflect other aspects of the models used for analyses. In contrast, it is straightforward to determine whether different analyses actually increase (or decrease) the congruence among the data subsets when analyses are conducted after dividing the data into subsets based on protein structure. Indeed, using this approach revealed at least one method able to increase congruence between the exposed and buried residues, at least for this dataset: conducting analyses after amino acid recoding.

### 4.2. Site-Heterogeneous CAT-Type Models do not Increase Congruence in Signal for Different Structural Classes

Site-heterogeneous models, especially the Bayesian implementation of the CAT model [15], have been used in many studies focused on deep branches in the metazoan tree (e.g., [37,38,87,88]). It has been asserted that CAT models are more biologically realistic than empirical models or the GTR model [15,39,89,90,91,92]. Analyses using CAT models have been suggested to be less prone to systematic error, such as long-branch attraction, than site-homogeneous models like the standard empirical models when they are applied to heterogeneous data [92,93,94,95]. The defining feature of CAT-type models is their use of a mixture of profiles to model amino acid propensities for different sites in proteins (see Section 2.5 in the Materials and Methods). The ability to model differences among sites in their amino acid propensities has been suggested to limit systematic error when analyses are conducted using CAT-type models [92,93].

Many studies from outside of the field of molecular phylogenetics corroborate the hypothesis that different sites in proteins have distinct propensities for amino acid occupancy. Site-to-site variation in amino acid propensities can explain the observation that databases searches using profiles based on multiple sequence alignments identify a larger number of distant homologs than in searches using a single query [85]. Deep mutagenesis studies have provided direct evidence that distinct sets of amino acids are necessary for function at different sites within proteins [72,96,97,98]. Of course, this assumes the assays used in deep mutagenesis studies are reasonable proxies for evolutionary fitness; however, that assumption is likely to be correct given that amino acid profiles generated by comparing homologous proteins are often similar to the those resulting from mutagenesis experiments (cf. Figure 5 in Melamed et al. [72]).

Despite the strong evidence that amino acid propensities vary across sites, it remains unclear whether CAT-type models actually provide the best framework for phylogenetic estimation. The CAT-type model seems reasonable from a theoretical standpoint if one postulates that the ***R*** matrix captures the rate at which novel mutations enter populations and that the profiles capture purifying selection by assigning disfavored amino acids at any given site very low equilibrium frequencies. However, changes in the site-specific amino acid propensities over time could lead to biases when such a model is used for analyses. The phenomenon of shifting amino acid propensities has been given a name: heteropecilly [99].

The potential for heteropecilly to bias phylogenetic estimation has been acknowledged [99,100]. It seems reasonable to postulate that analyses of heteropecillous data might be more problematic for CAT-type models with narrow profiles. Herein, we used empirical CAT-type models that we call LGL models based on the source of the profiles [67]. LGL models permit a test of the hypothesis that heteropecilly can bias phylogenetic analyses using CAT-type models; if heteropecilly is problematic LGL models with a smaller number of wide profiles (e.g., C10 and C20) should perform better than those with a larger number of narrow profiles (e.g., C50 and C60). Obviously, it is challenging to assess the performance of phylogenetic methods using empirical datasets because true phylogenies are unknown. However, we believe there are several criteria that can be used to assess the performance of phylogenetic methods in this study. First, there is a general criterion: analyses of two (or more) subsets of a dataset using better methods will yield the same topology (assuming both subsets are large enough to overcome stochastic error). Second, given a set of topologies with different *a priori* likelihoods of being correct, the tree that emerges in analyses of all subsets is more likely to be the one with the higher prior probability.

Judging the performance of various LGL models using our criteria resulted in a complex answer. Analyses of the exposed and buried subsets of the data using LGL models with larger numbers of profiles converge on the same topology (Table 3 and Table 4). Those results conflict with the prediction we made if heteropecilly is a source of bias. However, the analyses converge on topology T3 (Table 3 and Table 4); topologies T1 and T2 are more plausible than T3 (see the introduction and Section 4.4 of this discussion for details). Moreover, some LGL analyses yielded unexpected clades, like the sponge+placozoa clade and the clade uniting chordates and protostomes. As with the T3 topology, which emerged in analyses of exposed sites using LGL models with the largest number of profiles, deuterostome non-monophyly emerged in the LGL-BUR analyses with larger numbers of profiles (Figure 5). Taken as a whole, these results falsify the hypothesis that using CAT-type models to analyze subsets of the FRG dataset yield better estimates of phylogeny than single-matrix models (i.e., the GTR matrices estimated for each data subset). Moreover, our results indicate that, at least for the exposed and buried subsets of the FRG dataset, LGL models with larger numbers of profiles actually perform worse than those with fewer profiles.

The results indicating that increasing the number of profiles reduced the accuracy of phylogenetic estimation were surprising because the AIC*_c_* indicated that the C60 models had the best fit to the data (Table 3 and Table 4). This suggests the AIC*_c_* overestimates of the number substitutional categories appropriate for datasets; the zero or near zero estimated weights for specific substitutional profiles (Appendix A) provides another line of evidence for this idea. Of course, the Bayesian implementation of the CAT model, which uses the Dirichlet process prior to elicit profiles, could yield different results. However, we note that at least some other analyses using the Bayesian CAT model have recovered similarly troubling clades [42,101]. Moreover, the need to conduct the analyses in a Bayesian framework in order to use the Dirichlet process prior comes with a cost: the fact that Bayesian phylogenetic methods overstate clade support [102,103]. Artificially inflated support for weak topological conflicts (e.g., the conflicts evident in analyses of secondary structural elements; Figure 3) would have obscured the striking differences between exposed and buried sites.

The Bayesian CAT model was proposed at the onset of the “phylogenomic era” to facilitate analyses of the increasingly large sequence datasets becoming available at that time using a framework that accommodates heterogeneity among sites. Our results suggest that some CAT-type models are unable to capture some of the heterogeneity in the FRG data. Other ways to introduce heterogeneity to models of sequence evolution, like those using protein structure [16,17,104], might represent a better way to incorporate heterogeneity into phylogenetic analyses. Alternatively, single-matrix models (GTR, empirical models, or the Braun [105] physicochemical models) may be sufficient to accommodate the heterogeneity in large protein datasets once those datasets have been subdivided (or partitioned) in an appropriate manner. In this study, the conflict between trees that result from analyses of exposed and buried sites suggest that neither single matrix-models nor the CAT-type models we used can ameliorate the conflicting signals.

Ultimately, all models are approximations; the question of whether any specific approximating model is more likely than another approximating model to reveal the true historical signal is difficult to assess from theory alone and should also be examined empirically. We have presented a number of arguments that some CAT-type models actually behave worse than single-matrix models in analyses of the exposed and buried subsets of the FRG dataset. Obviously, these results raise general questions about the utility of CAT-type models, although we stress that we have only explored a small part of parameter space. Regardless of the details, our results certainly indicate that CAT-type models do not represent a panacea for phylogenetic analyses of proteins at this timescale.

### 4.3. Amino Acid Recoding Increases Congruence in Signal for Different Structural Classes

In addition to long-branch attraction, variation in amino acid (or base) composition across the tree is another source of systematic error in phylogenetic analyses [106,107]. However, the impact of variation in base composition on phylogenetic estimation is complex and difficult to predict. This complexity reflects the large number of ways that the proportions of the 20 amino acids can vary. We focused on the GARP/FYMINK ratio (Figure 6a), which correlated with genomic GC-content, because many other studies have found variation along the GC-AT axis [74,108,109]. Indeed, even in studies that have found significant variation along other axes in protein composition space (e.g., in a study focused on prokaryotes [110] an axis of variation correlated with optimal growth temperature and IVYWREL-content was evident) there appears to be much stronger variation on the GC-AT axis (in the Boussau et al. [93] study the GC-AT axis explained 45.4% of the variance among taxa whereas the IVYWREL-axis only explained 13.8%). Back translation of the amino acid sequences confirmed that there was more variation along the GC-AT axis than along the other two axes in nucleotide composition space (AC-GT and AG-CT; Figure 6b). The mean GARP/FYMINK ratios for exposed and buried sites were different, although taxa with an above average GARP/FYMINK ratio in one structural environment also had a higher ratio in the other environment (and vice versa). Thus, the relative pattern of variation in the GARP/FYMINK ratio among taxa did not differ between the structural environments. This makes it unlikely for variation among taxa in amino acid composition to represent the primary explanation for the observed differences in signal for the exposed and buried sites.

Despite the observed similarities in their patterns of compositional variation for exposed and buried sites, conducting analyses after recoding amino acids did increase congruence between exposed and buried sites (Figure 6c). In fact, analyses of exposed and buried data matrices generated by back translation with recoding the nucleotides as binary characters resulted in identical trees with relatively high (≥88%) bootstrap support for T2. We also conducted analyses after six-state recoding using the Dayhoff categories, an amino acid recoding method used in many studies (reviewed by Hernandez and Ryan [111]). Analyses of exposed sites after six-state recoding resulted in a topology identical to the tree generated by binary coding; support for T2 was also relatively high (90%). In contrast, analyses of buried sites after six-state recoding resulted in a tree with limited (<50%) bootstrap support for T2 and several other topological differences from the tree based on exposed sites after recoding. Indeed, many branches in the six-state recoded buried site tree had limited support (Appendix A). Regardless, it was clear that binary coding resulted in completely congruent trees for exposed and buried sites and six-state recoding increased congruence (albeit with high support only in the analysis of exposed sites).

Perhaps more important than the observed increase in congruence is the fact that amino acid recoding, especially binary coding, meets the criteria for a “well-behaved model” that we described above (in Section 4.2). Specifically, we argued that analyses of the exposed and buried subsets of this dataset using an ideal model would have three properties: (1) the trees generated using exposed and buried sites would exhibit greater congruence; (2) the congruent placement of the root would involve one of the more plausible topologies (T1 or T2 rather than T3); and 3) that unexpected (and relatively implausible) clades, like the sponge+placozoa clade, would not be recovered. Six-state recoding does not meet all of these criteria; analyses of buried sites recoded in this way yield the sponge+placozoa clade. However, support for that unexpected sponge+placozoa clade as well as the clade comprising all animals except ctenophores was very low when we conducted six-state recoding of the buried sites. Although it remains possible (in principle) that the behavior of six-state recoding would exhibit better behavior when used with a different (presumably larger) dataset it is clear based on our results that six-state recoding did not exhibit ideal behavior in this study. In contrast, binary coding meets all of the criteria described for a well-behaved analytical method.

The increased congruence observed when the data were recoded raises the question of whether the apparent improvements provide evidence that compositional variation explains the different signals evident in exposed and buried sites. The observation that the patterns of variation across taxa in the GARP/FYMINK ratio are quite similar for exposed and buried sites makes it unlikely that the differences in signal reflects a simple case of convergence in GC-content (which would be expected to lead to convergence either for GARP amino acids or for FYMINK amino acids). However, model violations due to changes in amino acid composition might have indirect effects on phylogenetic estimation, perhaps exacerbating other sources of bias (e.g., long-branch attraction). Alternatively, amino acid recoding could improve model fit in other ways that are difficult to assess. One important challenge for the use of amino acid recoding is that there are not, at this times, ways to assess model fit using objective criteria like the AIC*_c_*; Simmons [112] reviewed methods to choose the best way to encode data (e.g., as codons, amino acids, or even RY data) but statistical methods to choose among alternative amino acid alphabets do not exist at this point. Regardless of the details, it is clear that recoding amino acid data improves the congruence between estimates of phylogeny based on exposed vs. buried sites for the FRG dataset.

One relatively straightforward way that amino acid recoding might have a positive impact on phylogenetic estimation is by reducing the number of substitutions; this could reduce the potential for phenomena like long-branch attraction. However, reducing the number of substitutions also eliminates phylogenetic information. Indeed, Hernandez and Ryan [111] questioned the utility of six-state recoding, showing that the loss of phylogenetic information outweighs its benefits with respect to compositional heterogeneity. Our observation that analyses of the exposed sites results in relatively strong support after six-state recoding whereas analyses of the buried sites after six-state recoding results in very limited support is consistent with the hypothesis that six-state recoding results in substantial loss of information. This information loss is expected to be less problematic when the substitution rate is high; the higher rate of amino acid substitution in the solvent-exposed environment (note scale bars in Figure 2 and Figure 6c) is likely to make six-state recoding less problematic for the exposed sites. In contrast, binary coding preserves more information because most amino acids are recoded as two or three binary characters; we believe that binary coding of amino acids deserves more consideration in future studies.

### 4.4. Phylogenetic Implications

The primary goal for this study was to ask whether the signal revealed by phylogenetic analyses of a large-scale protein dataset was non-randomly distributed with respect to protein structure. We chose the RG dataset for several reasons, one of which was the fact that basal metazoan relationships had limited support when it was analyzed [6]. This suggested either that conflicting signals were present in the data or that there was very little signal (historical or non-historical) in the data. Our analyses revealed: (1) that conflicting signals were present in the data; and (2) that distinct signals were non-randomly distributed with respect to protein structure. However, those analyses did not establish which of those signals were historical in nature. This raises another fundamental question: what do our analyses reveal about the relationships among deep-branching metazoans?

One important feature of historical signal is that it is expected to be distributed broadly in the genome. Although there are certainly cases where true reticulations are present in the history of life [113,114], many relationships appear to reflect a “tree-like” history combined with discordance among individual gene trees superimposed on that tree for a variety of reasons (e.g., horizontal transfer and incomplete lineage sorting [25]). The structurally defined subsets of data that we examined represent different parts of the same 231 orthologous proteins, although the exact number of gene trees they represent is unclear because intra-locus recombination could lead to a case where one protein may be associated with multiple gene trees [115]. Regardless of the exact number of gene trees in our dataset, it is important to recognize that the exposed and buried sites are found throughout protein sequences, sometimes quite close to each other. Therefore, exposed and buried sites are encoded by sequences that are likely to represent virtually identical sets of gene trees. Thus, the historical signal present in both subsets of the data should reflect similar sets of gene trees so analyses of either subset of the data would be expected to converge on similar topologies (assuming the number of sites analyzed is sufficiently large). This was not what we observed for the majority of the analyses we conducted, but it was true for the analyses that used recoded amino acids. Those analyses converged on T2 (ctenophores sister to all other animals).

The observation that analyses of two non-overlapping phylogenomic datasets result in the same topology is not, in and of itself, sufficient to conclude that the tree topology in question (i.e., T2) is an accurate representation of evolutionary history. However, postulating that ctenophores are sister to all other animal does represent the simplest interpretation of the analyses we conducted; it is necessary to explain three different observations if one postulates that another topology is the best representation of evolutionary history. First, we observed that most phylogenetic analyses of exposed and buried sites in the FRG dataset yield distinct topologies (specifically, T2 and T3; e.g., Figure 2). This result is consistent with T2 and T3 but, as we have emphasized elsewhere, T3 is the least plausible of the three possible topologies. After all, T3 fails to explain the distribution of character states highlighted by Nielsen [35] any better than T2 but it has only been recovered in a subset of trees in a few phylogenomic studies [6,42]. Second, we found that the potential for the best-characterized sources of bias in phylogenetics (long-branch attraction and variation in amino acid composition) are likely to have a similar impact on analyses of exposed and buried site data (the relative branch lengths are similar for both structural environments [Figure 1 and Figure 6c] as are the patterns of variation in amino acid composition [Figure 6a]). Finally, we found that analyses of exposed and buried sites conducted after amino acid recoding converged on a single topology (T2). That final observation corroborates the hypothesis that T2 reflects evolutionary history.

Moving beyond the topology at the base of metazoa, a second difference was evident for many analyses of exposed and buried data. While analyses of exposed sites revealed a clade comprising placozoa and cnidaria (Table 1 and Figure 2) many analyses of buried sites placed placozoa sister to a cnidaria+bilateria clade (although support for the latter clade was low in a number of cases; Table 1 and Figure 2). Another recent study recovered placozoa+cnidaria clade [116] and it reported that the placozoa+cnidaria clade only emerged after excluding compositionally heterogeneous proteins. Perhaps surprisingly, Laumer et al. [116] also reported that most proteins exhibit evidence of compositional heterogeneity. Placozoa+cnidaria also emerged in a subsequent study with more extensive taxon sampling [117], but only after sites that appeared to exhibit compositional bias and/or saturation were excluded. A provocative aspect of our analyses is that they suggest several different analyses can yield the placozoa+cnidaria clade: (1) excluding sites and/or proteins that exhibit compositional biases [116,117]; (2) limiting consideration to residues on the surface of globular proteins; and (3) conducting analyses after RY-coding.

A third difference between trees resulted from analyses of exposed and buried sites was evident in the outgroup. Specifically, *Sphaeroforma* (the only member of Mesomycetozoea in the FRG dataset) was placed sister to Filozoa [82,118], a clade comprising Filasterea (represented herein by *Capsaspora*) and Apoikozoa (the choanozoa+metazoa clade), in analyses of exposed sites. In contrast, analyses of buried sites placed Apoikozoa sister to a *Capsaspora*+*Sphaeroforma* clade sister (Appendix A). Recent phylogenomic analyses support monophyly of Filozoa [119,120] and even the studies that are more equivocal regarding the relationship (e.g., Brown et al. [121]) recover monophyly of Filozoa in some analyses. As with our focal node at the base of metazoa and the placozoa+cnidaria clade, analyses of RY-coded data supported Filozoa monophyly (Figure 6).

Returning to the position of the metazoan root, we acknowledge that it remains possible to postulate that T1 (which sponges sister to all other animals) represents the correct position for the metazoan root, despite the observations described above. Obviously, one way to embrace the hypothesis that T1 is correct is to dismiss the FRG dataset as uninformative, perhaps due to the limited number of sites and/or taxa included. However, simply postulating that FRG dataset is uninformative makes it difficult to explain why the bootstrap support for T2 and T3 in analyses of the exposed and buried sites is actually higher than the support Ryan et al. [6] observed in analyses of the complete dataset.

Embracing T1 as the true topology while still acknowledging that the FRG dataset has the potential to be informative requires one to assume that our failure to recover T1 reflect bias. However, it is necessary to make several assumptions regarding the nature of the biases. First, one must postulate that most analyses of the exposed sites in the FRG dataset are biased toward T2. The exception to that bias toward T2 in analyses of exposed site would be the existence of a bias toward T3 evident in the subset of analyses using LGL models that yield T3 (Table 3 and Table 4). Second, one must assume analyses of the buried sites are generally biased toward T3. Finally, one must assume analyses of the exposed and buried subsets of the data conducted after amino acid recoding are both biased toward T2. Embracing all of those assumptions represents a much more elaborate hypothesis than simply postulating that T2 is correct and that the only biases are the support for T3 observed in analyses of the buried sites without recoding and in some analyses of exposed sites using LGL models. For that reason, we view our results as additional evidence corroborating the hypothesis that the root of the animal tree lies between ctenophores and all other metazoa.

### 4.5. Why Are Different Structural Environments Associated with Distinct Signals?

Our results raise a fundamental question: why do many analyses of the exposed and buried sites reveal different signals? Based on the arguments described above (in Section 4.4.) it is likely that the topology recovered in most analyses of exposed sites is the true metazoan tree (or, at the very least, the exposed topology is closer to the true tree). Thus, the question can be rephrased: why is the misleading signal in the FRG dataset disproportionately associated with buried sites?

The answer to this question is unlikely to be straightforward. The substitution rate is for buried sites is lower than the rate for exposed sited (note the scale bars in Figure 2 and Figure 6), so the association between misleading signal and sites with high substitution rates (which has been appreciated for over two decades; Bull et al. [122]), cannot answer the question. Likewise, differences in the overall amino composition for exposed and buried sites cannot explain the differences in signal because the amino acid frequencies are an intrinsic part of the models we used. Of course, the buried sites could have a smaller state space (average number of amino acids permitted at each site) than exposed sites; smaller state spaces will inflate homoplasy [123]. However, the average effective alphabet size for exposed and buried sites are quite similar (Appendix A), as is the amount of homoplasy associated with each class of sites (if the Figure 6c tree is correct, the retention index [124] is 0.3054 for buried sites and 0.3001 for exposed sites; Appendix A). These results exclude the relatively simple reasons for the observed difference between the two site classes.

Excluding simple explanations forces us to look to other fundamental difference between these site classes. One factor that might differ for sites that differ is the impact of epistasis, which likely to be a major source (if not the primary source) of heteropecilly. Fisher [125] defined epistatic interactions as any deviation between the observed phenotype for multiple mutations and the expected phenotype for a linear combination of those mutations. Defined in this way, epistasis has important implications for protein evolution (for review, see Starr and Thornton [126]) because that epistatic interactions causes amino acid propensities to change over time [73]. Put another way, epistatic interactions have the potential to cause heterpecilly. Could numbers of epistatic interactions (and the associated heterpecilly) represent the difference between exposed and buried sites?

There are good reasons to postulate that patterns of epistatic interactions for buried amino acids might exist for the two site classes. Buried amino acids have larger numbers of inter-residue contacts [49], making it reasonable to postulate that buried sites have the potential for a larger number of epistatic interactions than exposed sites. Deep mutagenesis has shown that pairs of mutations that exhibit epistatic interactions are more likely to be located close to each other [72], lending credence to this idea. Even if the average number of epistatic interactions is not the important factor distinguishing between the structural classes it remains possible that the types of interactions is important; intramolecular interactions are like to dominate for buried whereas epistasis involving two (or more) interacting molecules may be more important for surface residues. Epistatic interactions between residues involved in protein-protein interactions has been documented [127], but a full exploration of those interactions is currently very challenging. We acknowledge that these ideas are speculative, but the data necessary to test them are accumulating.

It is difficult to see a connection between epistasis and the observation that amino acid recoding (especially RY coding) improves analyses. By definition, amino acid recoding must act to reduce the impact of problematic site patterns on phylogenetic analyses, but there is no reason why it should have an impact on site patterns that reflect epistasis. On the other hand, if recoding is acting to limit the impact of compositional biases (a common justification for data recoding [9,14]), it might be acting indirectly (see Section 4.3 above). Ultimately, we uncertain whether the observation that amino acid recoding has a positive impact on phylogenetic analyses of the FRG dataset provide information about the reasons analyses of different site classes yield different topologies.

For this part of the discussion, we hypothesized that analyses of exposed sites performed better than analyses of buried sites; this suggests the true position of the metazoan root is T2 (ctenophores sister). It is more difficult to propose a hypothesis that can explain the distribution of misleading signal if T1 or T3 is assumed to be correct. If topology T2 excluded from consideration, postulating that T3 is correct and T1 is incorrect is simpler if one only considers the analyses conducted as part of this study. However, other lines of evidence indicate that T3 is the least plausible topology (see above, Section 4.4). On the other hand, we found no evidence for a signal that supports T1 so it would be necessary to postulate, as described above (Section 4.4), that a complex mixture of misleading signals are present in the FRG dataset and that those signals overwhelm any historical signal preserved in the dataset. Obviously, this makes it very difficult to devise a hypothesis able to explain the existence of those hypothetical misleading signals. Overall, the hypothesis that analyses of exposed sites reveal the true historical signal represents the simplest interpretation of our results. We expect tests of that hypothesis using other datasets to reveal additional information about processes that drive protein evolution and the topology for the deepest branches of the metazoan tree.

## 5. Conclusions

Our signal exploration corroborated the hypothesis that different phylogenetic signals are associated with specific parts of proteins that can be defined using non-phylogenetic criteria (in this case, structural criteria). Most analyses of data from two structural environments resulted in conflicting trees that each had relatively high support. Our analyses also suggest the use of site-homogeneous models (like GTR or empirical models such as LG) should not necessarily be eschewed in favor of CAT-type models like the LGL models, despite the observation that the latter models typically have a better fit to the data based on commonly used model selection criteria. This statement may appear to be problematic since it implicitly calls the use of standard model selection criteria into question, but we emphasize that it is actually very difficult to assess the fit of phylogenetic models to empirical dataset in absolute terms [128]. Even the most complex models currently available for phylogenetic analyses (including site-heterogeneous models), may be quite far from the true underlying processes of molecular evolution and, therefore, every bit as subject to systematic error as simpler models. Sanderson and Kim [129] worried that the very large AIC increases in response to the addition of relatively small numbers of free parameters might indicate that all models under consideration are very far from the (unknown) true model. Thus, the most parameter-rich models could be closer to the true model but still deviate from that true model in ways that actually obscure the historical signal. The available site-heterogeneous models might be very useful in some contexts, but they introduce many free parameters that are not necessarily constrained in a biologically realistic manner. Steel [130] worried that very parameter-rich models would be impractical because they would have to have enough parameters “to fit an elephant” (a colorful metaphor for an unrealistic and overly parameter-rich model that has been attributed to John von Neumann [131]). This “elephant factor” may explain the behavior of LGL models. One way to overcome this elephant factor is to place biological constraints on parameters; the association between protein structure and phylogenetic signal that we observed suggests protein structure could provide information regarding the best way to move forward when devising those constraints. Alternatively, it may be easier to identify models that reveal historical signal after simplifying the data in some way (e.g., by recoding amino acids). There has been limited exploration of the best way to select phylogenetic models when the data can be encoded in multiple ways, but it seems reasonable to state that further exploration of this approach is warranted. Regardless of the specific details, we believe that the identification of additional phylogenomic datasets where different data types are associated with distinct signals could provide a way to explore the behavior of phylogenetic methods and their ability to accurately recover true historical signal.

## Figures and Tables

**Figure 1 biology-09-00064-f001:**
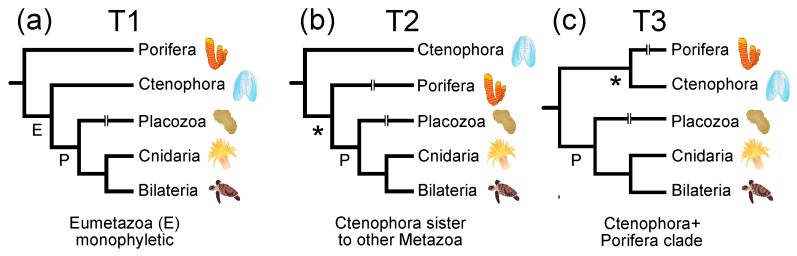
Topologies for the deepest branches in the metazoan tree recovered in phylogenomic analyses. (**a**) Porifera (sponges) sister to all other metazoa. This hypothesis includes a clade designated Eumetazoa (E). (**b**) Ctenophora (comb jellies) sister to all other metazoa. (**c**) A sponge+ctenophore clade sister to all other animals. All trees shown include a clade named Parahoxozoa (P) [43]. The Parahoxozoa topology was fixed based on King and Rokas [10], but an alternative topology with bilateria sister to a placozoa+cnidaria clade is also plausible.

**Figure 2 biology-09-00064-f002:**
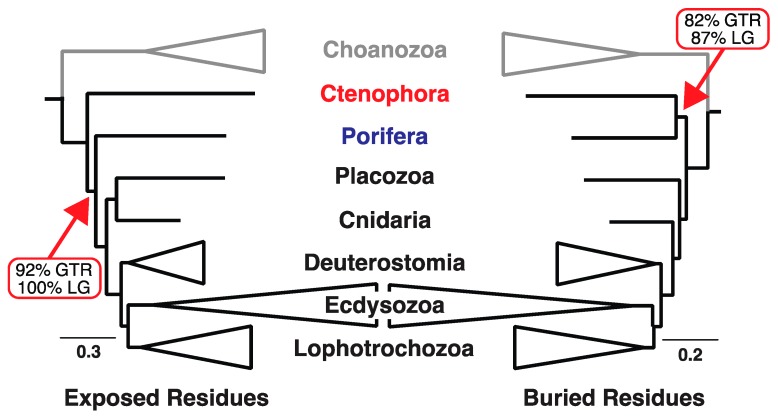
Analyses of sites from different structural environments reveal conflicting phylogenetic signals. We show simplified RaxML trees with both trees are limited to the metazoan ingroups and the choanozoan outgroup (i.e., only Apoikozoa *sensu* Budd and Jensen [78] are shown). The position of the root drawn in these trees was established by the outgroup taxa (the holozoans *Capsaspora* and *Sphaeroforma* and the fungi *Saccharomyces* and *Spizellomyces*). Bootstrap support for the positions of sponges and ctenophores given the general time reversible (GTR) model and the Le and Gascuel [52] (LG )model is indicated next to the arrow.

**Figure 3 biology-09-00064-f003:**
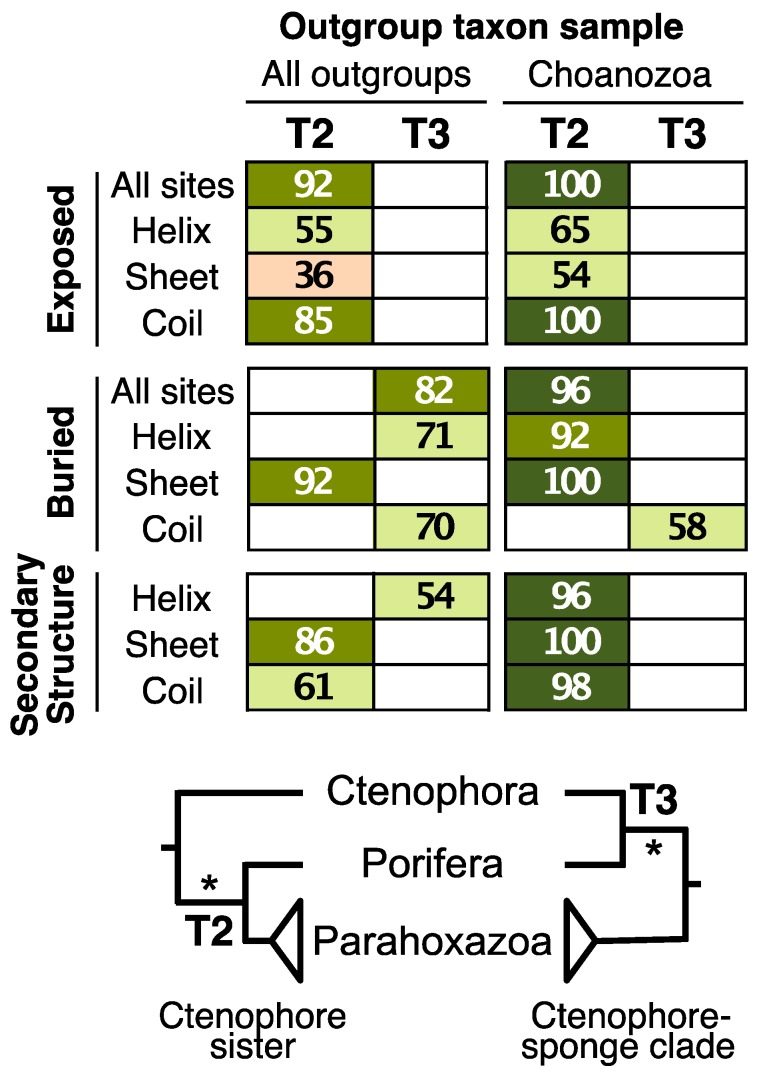
Heat map showing support for tree topologies obtained using various structural classes and taxon samples. In the online version colors indicate bootstrap support values (Dark green > 95, Lighter green >75, Yellow > 50 and Pink < 50; No color: Topology was not recovered).

**Figure 4 biology-09-00064-f004:**
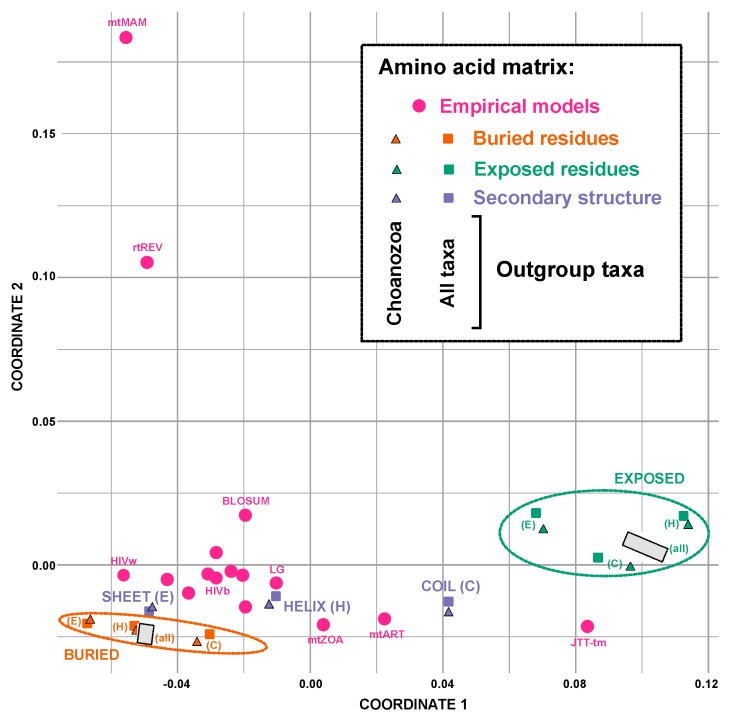
Multidimensional scaling plot showing the Euclidean distances between various amino acids exchange rate matrices. Different colors indicate different categories of the matrices in the online version (Green: exposed residues, Orange: buried residues, Purple: secondary structure, and Pink: standard empirical models).

**Figure 5 biology-09-00064-f005:**
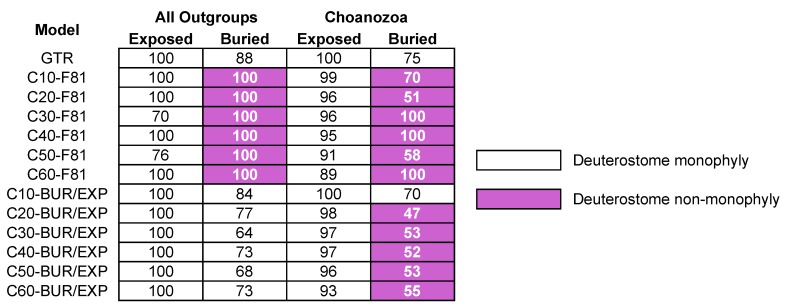
Heat map showing support for deuterostome monophyly for exposed and buried residues using GTR, LGL-F81, and LGL-BUR/EXP models. Colors indicate whether or not deuterostome were monophyly (No color: monophyletic, Purple: not monophyletic). Note that the bootstrap consensus tree for the LGL-BUR-C20 conflicted with the optimal tree; it had 53% support for monophyly.

**Figure 6 biology-09-00064-f006:**
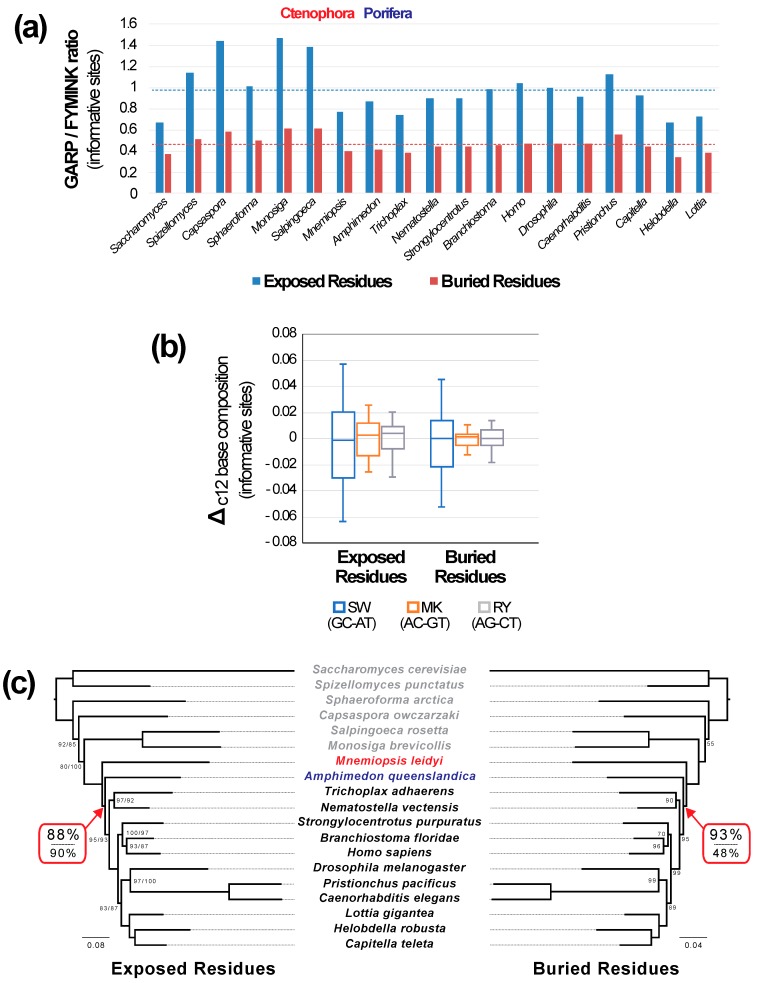
Compositional variation and the impact of recoding on tree estimation. (**a**) Variation across taxa in the ratio of amino acids encoded by GC-rich codons (G, A, R, and P) to those encoded by AT-rich codons (F, Y, M, I, N, and K). To limit the impact of invariant sites we only considered parsimony informative sites. (**b**) Variation in base composition for parsimony informative first and second codon positions after back translation. (**c**) The results of tree searches after recoding as binary (purine-pyrimidine (RY)) characters. The tree topologies were identical, although the tree lengths did differ (note scale bars). Support for the node that defines T2 (i.e., the node that places Ctenophora sister to other Metazoa) is emphasized using a red box; support given the binary data is presented to the top and the value for six-state Dayhoff recoding is presented below. For other nodes, bootstrap values <100% are presented, with values for six-state recoding presented to the right. Since the trees obtained after binary and six-state coding of buried sites exhibited some topological conflicts the bootstrap support for six-state coding is not presented on that tree (except for the focal node).

**Table 1 biology-09-00064-t001:** Log likelihood and AIC*_c_* values *^a^* for empirical models and the GTR model optimized for exposed and buried sites. The GTR model had the best fit (note bold AIC*_c_* value).

Structural Subset	Model	T2 *^b^*	T3 *^b^*	Cni+Bil *^c^*	Cni+Pla *^c^*	ln*L*	AIC*_c_*
Exposed	GTR	92	-	-	98	−1,212,232.985	**2,424,956.474**
	LG	100	-	-	100	−1,222,489.017	2,445,088.162
	WAG	87	-	-	100	−1,225,898.553	2,451,907.235
	VT	96	-	-	98	−1,225,672.633	2,451,455.395
	rtREV	90	-	-	98	−1,222,733.929	2,445,577.986
	JTTDCMUT	94	-	-	98	−1,229,028.451	2,458,167.031
Buried	GTR	-	82	64	-	−1,045,694.924	**2,091,880.072**
	LG	-	87	62	-	−1,050,022.577	2,100,155.267
	WAG	-	85	58	-	−1,054,829.256	2,109,768.626
	VT	-	81	61	-	−1,059,148.671	2,118,407.455
	rtREV	-	89	52	-	−1,054,432.123	2,108,974.359
	JTTDCMUT	-	81	-	54	−1,061,103.295	2,122,316.704

*^a^* AIC*_c_* was calculated using: AIC*_c_* = 2*k* − 2(ln*L*) + (2*k*^2^ + 2*k*) / (*n* − *k* − 1) where *k* is the number of parameters and *n* is the number of sites. This is the formula for the AIC*_c_* that is typically used in phylogenetics. *^b^* RaxML bootstrap support for T2 or T3. *^c^* RAxML bootstrap support for the cnidaria+bilateria clade or the cnidaria + placozoa clade.

**Table 2 biology-09-00064-t002:** Decisive sites favoring topology T2 vs. T3 in the exposed and buried residues *^a^*.

Site Class	Ctenophore Sister (T2)	Ctenophore + Porifera (T3)
Exposed	172	150
Buried	167	205

^*a*^ T2 and T3 refer to the arrangement of ctenophores, sponges, and remaining metazoa (Figure 1); other relationships were held constant.

**Table 3 biology-09-00064-t003:** Log likelihood and AIC*_c_* scores for various CAT-F81 models as implemented in IQ-TREE. The best-fit model is indicated by the bold AIC*_c_* value.

LGL-F81 Model	Outgroup	Dataset	T2	T3	Cni+Bil	Cni+Pla	Pori + Pla	ln*L*	AIC*_c_*

C10	All outgroups (RG)	Exposed	100			100		−1,225,103.471	2,450,339.126
C20		Exposed		63		100		−1,216,444.36	2,433,040.964
C30		Exposed	100				72	−1,214,319.593	2,428,811.499
C40		Exposed		100		96		−1,212,974.829	2,426,142.047
C50		Exposed	100				73	−1,212,360.091	2,424,932.657
C60		Exposed		100		93		−1,211,470.44	**2,423,173.448**
C10		Buried		83	78			−1,053,873.212	2,107,878.588
C20		Buried		97	80			−1,048,007.721	2,096,167.66
C30		Buried		94	73			−1,046,099.005	2,092,370.288
C40		Buried		92	83			−1,044,155.469	2,088,503.283
C50		Buried		93	86			−1,043,172.44	2,086,557.3
C60		Buried		93	89			−1,042,207.261	**2,084,647.025**

C10	Choanozoa only	Exposed	100			100		−964,491.9951	1,929,100.133
C20		Exposed	98			74		−957,706.2984	1,915,548.793
C30		Exposed	96			69		−956,161.8764	1,912,480.01
C40		Exposed	95			71		−955,297.444	1,910,771.215
C50		Exposed	92			51		−953,519.0992	1,907,234.604
C60		Exposed	90		56			−952,582.9203	**1,905,382.332**
C10		Buried		50	46			−837,871.4589	1,675,859.045
C20		Buried		37	34			−833,463.787	1,667,063.748
C30		Buried		75	63			−832,000.2662	1,664,156.761
C40		Buried		66	63			−830,714.1456	1,661,604.582
C50		Buried		45	45			−829,970.749	1,660,137.858
C60		Buried		74	79			−829,121.6382	**1,658,459.713**

**Table 4 biology-09-00064-t004:** Log likelihood and AIC*_c_* scores for various LGL-BUR and LGL-EXP models. The best-fit model is indicated in bold and the FO mixture weight is reported.

LGL Model	Outgroup	Dataset	T2	T3	Cni+Pla	FO Weight	ln*L*	AIC*_c_*

EXP-C10	All outgroups (RG)	Exposed	100		100	0.2774	−1,209,875.9003	2,419,883.9851
EXP-C20		Exposed	100		99	0.1808	−1,206,472.3633	2,413,096.9708
EXP-C30		Exposed	100		99	0.126	−1,204,598.8783	2,409,370.0689
EXP-C40		Exposed	100		99	0.105	−1,203,943.3823	2,408,079.1535
EXP-C50		Exposed	100		99	0.1139	−1,204,270.9338	2,408,754.3413
EXP-C60		Exposed		63	100	0.1146	−1,203,752.4217	**2,407,737.4104**
BUR-C10		Buried		93	73	0.3637	−1,043,644.2586	2,087,420.6812
BUR-C20		Buried		95	81	0.1584	−1,037,855.6842	2,075,863.5853
BUR-C30		Buried		95	84	0.0741	−1,037,721.2711	2,075,614.8197
BUR-C40		Buried		91	86	0.0797	−1,036,144.9852	2,072,482.3158
BUR-C50		Buried		95	79	0.0806	−1,035,169.3326	2,070,551.0859
BUR-C60		Buried		95	84	0.0892	−1,034,522.1735	**2,069,276.8507**

EXP-C10	Choanozoa only	Exposed	92		97	0.2551	−952,094.7392	1,904,305.6212
EXP-C20		Exposed	87		97	0.176	−949,973.8488	1,900,083.8935
EXP-C30		Exposed	77		91	0.0604	−948,352.7546	1,896,861.7664
EXP-C40		Exposed	77		98	0.0629	−947,967.3623	1,896,111.0517
EXP-C50		Exposed	76		95	0.0711	−948,154.1412	1,896,504.6876
EXP-C60		Exposed	72		96	0.0566	−947,765.6493	1,895,747.7905
BUR-C10		Buried	67		96	0.2376	−826,008.6321	1,652,133.391
BUR-C20		Buried	48 *^a^*	42 *^a^*	93	0.1838	−822,299.4844	1,644,735.1427
BUR-C30		Buried	74		97	0.0981	−821,989.872	1,644,135.9726
BUR-C40		Buried	38 *^a^*	37 *^a^*	96	0.1211	−820,830.2779	1,641,836.8464
BUR-C50		Buried		74	96	0.0950	−820,251.2702	1,640,698.9004
BUR-C60		Buried		78	91	0.1125	−819,722.1126	1,639,660.6619

*^a^* The optimal tree and bootstrap consensus trees differ for these LGL-BUR analyses; the optimal tree was T3 and the bootstrap consensus was T2.

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
