# Peer review of "Phylogenetic Analyses of Sites in Different Protein Structural Environments Result in Distinct Placements of the Metazoan Root"

_biology, 2020, doi:10.3390/biology9040064_

Round 1

Reviewer 1 Report

This paper entitled “Phylogenetic Analyses of Sites in Different Protein Structural Environments Result in Distinct Placements of the Metazoan Root” by Pandey Braun examines the phylogenetic signal contained in two structurally defined subsets of a phylogenomic dataset as it relates to the Metazoan phylogeny. The authors find that phylogenomic datasets comprised of stretches of solvent-exposed residues support ctenophores as the sister to the remaining metazoan under a range of models, while the dataset derived from buried residues supports a monophyletic clade comprised of ctenophores plus sponges. This latter result is dependent on model selection and varies between alternative models.

The paper is well written and provides comprehensive analyses. While I do have some critiques of the manuscript, they are largely minor and the paper stands as a reasonable contribution to the field of functional phylogenomics.

Essential revisions:

The authors discuss two findings, deuterostome polyphyly and a monophyletic clade of ctenophores plus sponges, which were both previously reported in recoding treatments of a comparable phylogenomic dataset by Borowiec et al (2015; Figure S4), but this paper is not cited.

https://bmcgenomics.biomedcentral.com/articles/10.1186/s12864-015-2146-4

In addition, the present manuscript reports the finding of a monophyletic placozoa plus cnidaria which has also been previously reported by Laumer et al 2018).

https://elifesciences.org/articles/36278

The manuscript should site the previous works.

Line 129:  I think a word is missing in “differed from associated with buried residues”

Optional revisions:

The authors find what could be an important result, that phylogenetic signal differs between structurally defined datasets, but they never offer a biological explanation for why this might be? The authors should feel free to speculate on a molecular evolutionary explanation for this result.

The authors focus on a single phylogenomic dataset but never justify why the Ryan dataset was chosen. Several other, larger, more comprehensive phylogenomic datasets for metazoans are now available. Why was the Ryan dataset chosen? Could this the central finding of this paper be a peculiarity of the Ryan dataset that was chosen?  

Author Response

Thank you very much for your valuable comment. We have provided our responses in the attached file.

Author Response

Thank you very much for your valuable comments. We have provided our responses in the attached file.

Round 2

Reviewer 2 Report

General comments:

I thank the authors for providing detailed responses to my first review, and for making considerable edits to the manuscript. In may ways, this is a substantially improved paper. I especially like the use of the C10-C60-GTR models. There are some grammatical issues in revised text sections (see below, but I could have missed some) so please give a careful edit to the revised sections.

Broadly, I think there are still two major issues that must be fixed:

The terminology used to refer to the C10-C60 models is unacceptable. First, I apologize for asserting in my first review that the authors did not seem to understand the differences between various site-heterogeneous models as they clearly do. That said, I can see no justification for referring to the C10-C60 models as CAT models without qualification, particularly in the abstract. It is extremely confusing. I do not understand why the authors opted for confusion rather than clarity in word choice. Please see below for more details. The authors need to better articulate the goal of their paper and include conclusions with appropriate caveats. If the goal was to show that different structural classes can have different signals, then I’m not sure how universal the observed phenomenon is given that one dataset was analyzed. If the goal was to show how different structural protein classes could affect metazoan relationships, as the title implies, then the dataset analyzed was not appropriate, as taxon sampling is so low. I have to wonder if it is too much to ask to analyze at least one other dataset? Please see below.

Terminology for site-heterogenous models: I apologize to the authors as they clearly understand the models chosen, and my first review inelegantly suggested they did not. That said, I cannot see any reason to not be more explicit about the specific models used, especially in the abstract. If the authors want to say something like “using the maximum likelihood C10-C60 site-heterogeneous models implemented in IQTREE, herein referred to as CAT models” then I guess that could be OK. However, the current wording is very unclear as the maximum likelihood CAT model could be mistaken for the CAT approximation of modeling rate-heterogeneity in RAxML. While there are certainly similarities between the CAT models in PhyloBayes and the C10-C60 models, they are not the same models. Moreover, they have been shown to perform differently in respect to the metazoan root (see Fueda et al. 2107). The way the paper refers to CAT models is horribly confusing, going as far as referencing different papers to refer to different models that the authors refer to as CAT models with no distinction. I think the use of the C10-C60 models is fine for this study, and I think the authors are right that analyses with the CAT models in phylobayes are outside the scope of the current study. Nevertheless, the authors need to be clear about which models they are talking about and used before the paper can be published.

The use of “CAT-F81” and “CAT-GTR” to describe the C10-C60 models in lines 330-338 is not acceptable. I’m aware of no study that has ever called the C10-C60 models CAT-F81 or CAT-GTR, and the field generally considers CAT-GTR and CAT-F81 as the CAT models implemented in PhyloBayes. There is no reasonable justification for confusing the terminology as done in this paper, unless the goal is to not be clear.

What’s most frustrating is that this semantic issue obfuscates the fact that the authors analyzed the dataset with a C10-C60-GTR model (of sorts), which may be a particularly useful way to employ the C10-C60 models. I am not aware of a paper that has used the C10-C60 models in this way, and I think it is interesting.

Dataset choice

I still have concerns about the dataset that was chosen for this study. If the primary goal, as the authors state in the response document, was to establish whether different sites in different structural environments were associated with distinct phylogenetic signals then I think analyzing more datasets would have been appropriate. Essentially, they have shown in one dataset that there is a difference of signal. While interesting, it is hard to make conclusions about the true tree or if this is a problem phylogeneticists must be concerned with for accurate phylogenetic inference. The authors acknowledge this in the discussion. On the other hand, the paper spends a lot of time discussing metazoan relationships, so it is hard for me to disentangle the stated primary goal with a major focus of the paper. The title even emphasizes the placement of the metazoan root. For that reason, I think more taxon-rich datasets would have been appropriate for a study that focuses so much on the placement of the metazoan root.

To the authors point about difficulty in finding datasets with a conflicting signal: I think this is pure speculation. The authors acknowledge they have only analyzed one dataset, so they do not know if other datasets would have this issue. Perhaps the “problem” with conflicting signal is alleviated by increased taxon sampling. Or, perhaps datasets with only highly supported nodes also have conflicting signals that confound phylogenetics. I do not know answers to these questions, but either datasets with other focal taxa should be analyzed to identify if conflicting signal is a universal phenomenon or a more taxon rich dataset should be analyzed if the position of the metazoan root is a main focus of the study.

At the end of the day, I find the results somewhat interesting. However, the paper would definitely be improved if another dataset was analyzed.

Line by line comments

Line 87: I’m not too swayed by this argument for dataset choice. Numerous papers have demonstrated that a small number of genes can have strong conflicting signal that is not apparent when the dataset is analyzed in total (e.g. https://www.ncbi.nlm.nih.gov/pmc/articles/PMC5560076/).

Lines 89-91: This sentence is confusing and needs editing.

Lines 161-163: Or it could be both.

Lines 271-287: I think this was a good analysis, and I thank the authors for adding it. However, the authors should be more explicit that these methods appeared to work well for masked alignments, which seems to be glossed over.

Lines 197-206: Why report analyses with lesser-fitting models? Does that not mean that analyses with known sources of systematic error are being reported? I also cannot find anywhere in the methods where information about model testing, as reported in Table 1 and 3, was done. I also think partitioning schemes within the structural classes should have been examined. Coupled with my concerns about dataset choice, the authors have simply not convinced me that their results are not a result of limited taxon sampling and/or less than idea model choice.

Line 447: Please remove “would”.

Line 602: Change “separated” to “separating”.

Line 604: I don’t think a statistical test was done to measure significance, so this sentence should be reworded. Or a p-value should be reported.

Lines 670-672: Does increased congruence really equal increased accuracy? I do not think so. Amino acid recoding results in the loss of a lot of information, which could explain the increased congruence but not the increased accuracy. This sentence needs to be better qualified, especially in light of Hernandez and Ryan and lines 826-839 in this paper.

Lines 928-929: “that the” is repeated.

Round 3

Reviewer 2 Report

General Comments

I’m disappointed the authors did not examine at least one other dataset. As I previously expressed, I felt strongly that the results could be a byproduct of analyzing a subset of a taxon-poor dataset, rather than a result of truly different signal in protein structural classes.

As a test of this hunch, I twice created random subsets of the Ryan et al. dataset in approximate proportions to the amount of data in each subset of the current study (i.e., buried vs exposed sites). This resulted in subsets of 53% of genes and 47% of genes randomly selected without replacement. I analyzed these datasets in IQTREE under the LG+F+G model, which is equivalent to the RAxML models used for the author’s LG analyses. The inferred trees differed in similar ways to the trees inferred in the paper. That is, in subset 1, the 47% dataset had ctenophores sister to all other animals and the 53% dataset had sponges sister to all other animals. In subset 2, the 53% dataset had sponges + Ctenophora sister and the 47% dataset had ctenophores sister to all other animal. That is even random splitting of the data in proportions similar to the percentages of buried and exposed sites in the RG creates datasets with different signals. If randomly sub-setting the Ryan et al dataset can result in two datasets with different signals and inferred trees that are nearly identical to those inferred by the authors, then the authors cannot say that the patterns observed in their analyses have anything to do with different protein structural classes.

As I stated before, if the authors want to test that buried vs exposed sites have truly different signals, rather than smaller datasets having more noise, they need to analyze other datasets. If the authors refuse to do this, then I cannot recommend this paper for publication. Many (almost all?) studies that examine performance of dataset types and models in a phylogenetic context examine more than one dataset; without doing so, generalizations would be impossible to make about model or dataset performance. My quick analysis of random subsets demonstrates why it’s problematic to 1) analyze a dataset with such limited taxon sampling, 2) make broad generalizations (i.e., buried vs exposed sites have different signals) based on a single dataset.

I acknowledge that the above analysis is not very thorough, but it does provide more than enough reason for requiring additional datasets to be analyzed before this study can be considered acceptable for publication.

I have uploaded a zipped file of the datasets. Subset 1 refers to the first random splitting with 53% of genes in subset1_53 and the remaining 47% of genes being in subset1_47. Subset2 is structured the same way, but was created with a different random sampling of genes.   

I have not made line-by-line comments because more datasets would need to be analyzed for a future version of this study to be published. I think this would require a rewrite of the paper.

Finally, I apologize for citing the wrong paper in a past review (Fueda et al) when I should have cited Simion et al. (2017) as the paper that demonstrated performance differences between C10-C60 models and CAT models.
